# Complete Mitochondrial Genome, Genetic Diversity and Phylogenetic Analysis of Pingpu Yellow Chicken (*Gallus gallus*)

**DOI:** 10.3390/ani12213037

**Published:** 2022-11-04

**Authors:** Sihua Jin, Jingjing Xia, Fumin Jia, Lijun Jiang, Xin Wang, Xuling Liu, Xing Liu, Zhaoyu Geng

**Affiliations:** 1College of Animal Science and Technology, Anhui Agricultural University, Hefei 230036, China; 2Anhui Provincial Key Laboratory of Local Animal Genetic Resources Conservation and Bio-Breeding, Hefei 230036, China

**Keywords:** Pingpu Yellow chicken, mtDNA, phylogeny, D-loop region

## Abstract

**Simple Summary:**

Characterization of the complete mitochondrial genome can provide information regarding important genomic features, such as genome size and the relative location of genes, and is widely used in studies on genetic diversity and phylogeny of animals. The displacement loop region (D-loop) of mitochondrial DNA (mtDNA) is considered to be a beneficial molecular marker for genetic distance determination and differentiation analysis. This study presents the mtDNA diversity and genetic characterization of Pingpu Yellow chicken (PYC) breeds, including insights into their genetic background. The median-joining (MJ) network showed that the haplotypes were clustered into all three haplogroups (A, B, and C), indicating that PYC may have three maternal origins. The phylogenetic tree constructed with the neighbor joining (NJ) method based on the full mitogenomes showed that the 21 chicken breeds could be separated into two groups.

**Abstract:**

In this study, the complete mitochondrial genome sequence of one female Pingpu Yellow chicken (PYC) and the D-loop sequences obtained from 60 chickens were analyzed to investigate their genetic diversity and phylogeny. The total length of the PYC mitogenome is 16,785 bp and that of the complete D-loop is 1231 to 1232 bp. The mitogenome comprises 22 transfer ribonucleic acids (tRNAs), 2 ribosomal ribonucleic acids (rRNAs), 13 protein-coding genes (PCGs), and 1 non-coding control region (D-loop). Additionally, the total length of the 13 PCGs is 11,394 bp, accounting for 67.88% of the complete mitogenome sequence, and the PCGs region has 3798 codons. A majority of the PCGs have ATG as the start codon. The haplotype and nucleotide diversity of PYC were 1.00000 ± 0.00029 and 0.32678 ± 0.29756, respectively. In the D-Loop data set, we found 25 polymorphic sites, which determined 18 haplotypes and 3 major haplogroups (A–C). Therefore, PYC has a classical vertebrate mitogenome, with comparatively high nucleotide diversity and potentially three maternal lineages. The neighbor-joining (NJ) tree analysis results showed PYC grouped with the Luhua (MT555049.1) and Nandan chickens (KP269069.1), which indicates that PYC is closely related to these two breeds.

## 1. Introduction

Mitochondrial DNA (mtDNA) is maternally inherited, and the highly variable D-loop region of its sequence is often used for phylogenetic and phylogeographic analyses. Several mtDNA sequences have been used in phylogenetic analyses, including 12S rRNA, 16S rRNA, CytB, and COI genes [1,2]. Analysis of the D-loop region was successfully applied in an investigation of the origins and genetic diversity of Chinese domestic chickens. mtDNA sequence polymorphisms have also been used to investigate maternal origins, domestication events [3,4,5], and genetic relationships within and among European and African chicken populations [6,7].

The domestication of chickens is traced to 2000 BC. Chickens evolved from red jungle fowl (*Gallus gallus*) under the influence of natural and artificial selection [8,9]. The Pingpu Yellow chicken (PYC), a medium-sized breed, is a popular local chicken of the Anhui province in China (Figure 1). Currently, researchers have utilized the full-length mtDNA D-loop sequence in population analysis of chicken [10,11,12], sheep [13,14], and cattle [15,16]. In this study, using complete and partial mtDNA sequences, we assessed the phylogenetic relationship between PYC and several related breeds. We extracted DNA from the blood of adult birds and sequenced the complete control region. Consequently, we analyzed the genetic diversity and carried out the phylogenetic characterization of PYC. Our results may provide insights into the genetic background of PYC and a basis for further research on its evolution.

## 2. Materials and Methods

### 2.1. Ethics Statement

Chickens were acquired from Wuhu Zhongs Poultry Conservation and Breeding, Co. Ltd., Anhui, China. All protocols involving the chickens were approved by the Animal Care and Use Committee of Anhui Agricultural University (approval number: SYXK 2016-007).

### 2.2. Specimen Collection and DNA Extraction

We randomly selected PYCs with complete pedigree information. Blood samples from 60 adult chickens (30 males and 30 females) were obtained from the farm, the total genomic DNA was isolated using a DNA extraction kit (Tiangen, Beijing, China), and stored at −20 °C. DNA integrity was assessed using 1.5% agarose gel electrophoresis. One female PYC was randomly selected to study the complete mitochondrial genome by next-generation sequencing, and a total of 60 samples were used to determine the D-loop sequences.

### 2.3. PCR Amplification and DNA Sequencing of D-Loop

Using the full mitochondrial genomic data of *Gallus gallus* (NCBI, GenBank: GU261704.1), the length of the D-loop was determined to be 1 231 bp. Oligo7 software [17] was used to design the primers (F: 5′-CAAACTCACTAACCACCCA-3′, R: 5′-GCCTGATACCTGCTCC-3′) for the D-loop, and these were synthesized by Nanjing Kinco Biotechnology (Nanjing, China). PCR was performed using a 25 μL reaction mixture containing 9.5 μL ddH_2_O, 12.5 μL 2Taq PCR Master Mix, 1 μL (50–100 ng) template DNA, and 1 μL of each primer. The cycling conditions were 94 °C for 5 min followed by 1 cycle of 94 °C for 30 s, 60 °C for 30 s, 72 °C for 60 s, and a final extension step of 10 min at 72 °C. The specificity of the primers was assessed by performing a 1.5% agarose gel electrophoresis of the PCR products. A pair of primers was selected as the D-loop amplification primers in the present study.

### 2.4. Next-Generation Sequencing of the Mitogenome

The DNA was fragmented using ultrasound, and the fragmented DNA was purified and selected by 1.5% agarose gel electrophoresis. PCR amplification was performed to create a sequencing library, which was then subjected to library quality control. Then, the qualified library was sequenced on an Illumina NovaSeq 6000 platform, with paired-end read lengths of 150 bp. 

### 2.5. Data Analysis

After data filtering, the whole mitogenome sequence was edited and aligned with the reference genome (GU261704.1) by Spades software (Version: 3.13.0; parameter: -K 127). The splicing results were compared by BLASTN (Version: BLAST 2.2.30+; parameters: -evalue 1 × 10^−5^), and the candidate sequence assembly results were determined based on the alignment. We used MITOS2 (http://mitos2.bioinf.uni-leipzig.de/index.py, accessed on 16 August 2021) for gene annotation. The circular mitogenome map was depicted using OGDRAW [18]. The nucleotide composition and relative synonymous codon usages (RSCU) of PCGs of PYC were calculated using MEGA v7 [19]. The strand asymmetries were computed using a formula, AT-skew = (A − T)/(A + T) and GC-skew = (G − C)/(G + C) [20]. The secondary structure of tRNAs was inferred by the tRNAscan-SE v2.0 [21]. We downloaded 20 chicken breed reference sequences from NCBI (GenBank: KP681581.1 Guangxi three-buff, KM433666.1 Cenxi three-buff, MH732978.1 Lindian, KX987152.1 Zhengyang yellow, MK163565.1 Chahua, KY054997.1 Lueyang yellow, KJ778617.1 Taihe, KF981434.1 Taoyuan, GU261719.1 Chigulu, GU261678.1 Gushi, GU261676.1 Wuding, GU261713.1 Jiangbian, KX781318.1 Zhuxiang, KF954727.1 Huang Lang, KP244335.1 Hengshan yellow, KF826490.1 Xuefeng black boned, MK163562.1 Tibetan, GU261677.1 Xianju, MT555049.1 Luhua, and KP269069.1 Nandan chickens). Molecular phylogenetic analyses and visualization of the neighbor-joining (NJ) tree were conducted using MEGA v7 software [19,22]. The evolutionary distances were computed by the Kimura 2-parameter (K2P) model [23], and corresponded to the number of base substitutions per site.

D-loop sequences were assembled and aligned using Contig Express in Vector NTI Advance 11 sequence analysis software (Invitrogen, Waltham, MA, USA) [24] against the reference sequence of *G. gallus* mtDNA (GU261704.1) using CLUSTAL software [25]. DNAsp v6 software [26] was used to analyze the number of PYC D-loop haplotypes, haplotype polymorphism sites, and nucleotide polymorphisms. Some haplotype sequences for chickens [27,28] were downloaded from NCBI (GenBank: A1, GU261684; A2, GU261695; A3, GU261700; B1, GU261704; B2, GU261705; B3, GU261714; B4, GU261699; C1, GU261681; C2, GU261718; C3, GU261716; D1, NC_007236; D2, GU261683; D3, GU261677; E1, GU261713; E2, GU261712; E3, GU261694; F, GU261711; G, GU261676; H, GU261715; X, GU261692; W, GU261706; Y, GU261693; Z1, GU261674; Z2, GU261696), and NETWORK10.2 software (Fluxus Technology Ltd., at www.fluxus-engineering.com, Cambridge, United Kingdom, accessed on 18 October 2021) was used to draw a haplotype network diagram.

## 3. Results

### 3.1. The Analysis of Complete mtDNA 

The size of the PYC mitogenome is similar to that of the mitogenomes of other chicken breeds. The complete mtDNA length of PYC is 16 785 bp (GenBank accession number: MZ911748). Characterization of the mitogenome revealed thirteen PCGs, seven of which correlate with the diverse subunits of NADH (NADH1-6 and NADH4L), three with COX1-3, two with ATP synthase subunits (ATP6 and ATP8), and one with cytochrome b (CytB). The mitogenome also consists of 2 rRNA genes (rRNAL and rRNAS) and 22 tRNA genes. Eight tRNA genes, ND6, and two rRNA genes are situated on the negative strand, and the remaining 14 tRNA genes and 13 PCGs on the positive strand (Table 1).

According to the mtDNA genome map, the base composition was A: 30.3%, T: 23.7%, G: 13.5%, and C: 32.5%. The GC content (46.0%) is lower than the AT content (54.0%) (Figure 2). 

### 3.2. Codon Usage and PCGs

The length of 13 PCGs in PYC is 11,394 bp, accounting for 67.88% of the whole mitogenome sequence. The 13 PCGs ranged in length from 165 bp (*ATP8*) to 1 818 bp (*ND5*). The PCGs region of the PYC mitogenome consists of 3798 codons. Most of the PCGs have ATG as a start codon, except for that of *COX1*, which uses GTG as the start codon. The longest coding sequence (CDS) is *ND5* (1818 bp), which is located between *ND4* and *CYTB*. The shortest was *ATP8* (165 bp), which is located between *COX2* and *ATP6*. The standard stop codon TAA occurs in *ND1*, *COX2*, *ATP6*, *ATP8*, *ND3*, *ND4L*, *ND5*, *CYTB*, and *ND6*; *COX1* and *ND4* use AGG and TGA as the stop codon (Table 2 and Figure 3). 

The codon usage in PCGs is biased toward amino acids encoded by A-rich codons (two or more A’s per triplet). The most frequently used codons are all A-rich, with the most common codons being TAA (34.62%) and AGA (3.84%). In contrast, the frequency of G-rich codons (two or more G’s in a triplet) was 11.54%.

### 3.3. Transfer and Ribosomal RNA Genes of PYC

Most of the tRNAs could be folded into the canonical cloverleaf secondary structure, except for tRNA-Ser (GCT), which lacks the DHU arm (Figure 4). Regarding the genetic diversity and population structure, we found that the nucleotide diversity (Pi) was 0.00600 ± 0.00451, and haplotype diversity (Hd) was 0.93600 ± 0.00020. The tRNAs ranged in size from 66 bp (trnC) to 76 bp (trnW).

The whole length of two rRNAs was 2600 bp. The AT content occupied 60.1%, AT-skew was positive (0.2360), and GC-skew was negative (−0.2189).

### 3.4. Structural Features and Sequence Composition of the Full-Length mtDNA D-Loop

The whole mtDNA D-loop succession was in the range of 1231 and 1232 bp. Most (66.33%) of the sequences were from the 1231 bp haplotype. The base composition is 30.3% A, 23.7% T, 13.5% G, and 32.5% C. The GC content (46.0%) is lower than the AT content (54.0%). A total of 25 polymorphic sites and 18 haplotypes were identified and named sequentially from PYC1 to PYC18 (GenBank accession numbers: MZ971232 to MZ971249). Among the haplotypes, three groups could be distinguished (Table 3).

The first group (haplogroup A) consisted of twelve haplotypes (PYC1, PYC2, PYC3, PYC4, PYC5, PYC7, PYC10, PYC11, PYC12, PYC13, PYC16, and PYC17); the second group (haplogroup B) consisted of three haplotypes (PYC6, PYC8, and PYC9); and the third group (haplogroup C) consisted of three haplotypes (PYC14, PYC15, and PYC18) (Figure 5). Transitions and substitution mutations were the cause of all polymorphic locations. We used NETWORK 10.2 software (Fluxus Technology Ltd., Cambridge, UK, at www.fluxus-engineering.com, accessed on 18 October 2021) to compare the 18 haplotypes of PYC with haplogroups A, B, and C [27,28] and found that the haplotypes were clustered into all three haplogroups (A, B, and C), although mostly in haplogroup A, indicating that PYC may have three maternal origins.

The circular areas correspond to the haplotype frequency. Yellow circles are haplotypes determined in this study. Blue circles are 24 haplotypes which were found previously.

### 3.5. Phylogenetic Reconstructions Based on Complete mtDNA

The phylogenetic tree compared PYC with 20 reference breeds. PYC was found to be in a cluster with the Luhua (MT555049.1) and Nandan (KP269069.1) chickens, based on the results of the neighbor-joining tree analysis, which indicates that PYC is closely related to these two breeds (Figure 6).

## 4. Discussion

There is little recombination in the mitochondrial genome, which is only passed down from mother to child [29]; however, nucleotide substitution rates of the mitochondrial genome are five to ten times higher than those of the single-copy genes in the nucleus [30]. The vast majority of mtDNA mutations have very few insertions or deletions. Additionally, owing to the various rates of mitochondrial gene evolution, various genes in the mitochondrial genome can be utilized in different phylogenetic and population genetic studies [31]. Furthermore, mitochondrial genes are easier to identify because of their close association. Over the past four decades, mitochondrial genes have been widely used as phylogenetic molecular markers. In studies dealing with conservation genetics, a plethora of genetic markers serve as useful tools. However, the variety of markers that can be used is limited by the difficulty of obtaining high-quality DNA from PYC populations present in the wild. Therefore, to investigate the PYC genetic variation, we used mtDNA control region (CR) sequences. Our results suggest that the CR sequence in PYC has the same structure as that of other vertebrates [32,33]. 

The length of the PYC mtDNA was the same as that reported by Gu and Li [34] and Miao et al. [28]. Thus, this study provides baseline genetic information for PYC and used mitochondrial markers to determine maternal origin. mtDNA analysis revealed three major matrilineal origins: clades A, B, and C. These findings demonstrate a genetic association between PYC and the Luhua (MT555049.1) and Nandan (KP269069.1) chicken breeds; thus, indicating that PYC is closely related to them.

## 5. Conclusions

The sequencing and annotation of the mitogenome of *G. gallus* and its comparison with the mitogenomes of other chicken breeds showed that the mitogenome characteristics of *G. gallus* are mostly consistent with those of the other chicken mitogenomes that have been reported. PYC genetic diversity, population structure, and phylogeny can be assessed by analyzing the full-length D-loop sequence of its mtDNA. The findings of this study will have useful application for the breeding and conservation of PYC.

## Figures and Tables

**Figure 1 animals-12-03037-f001:**
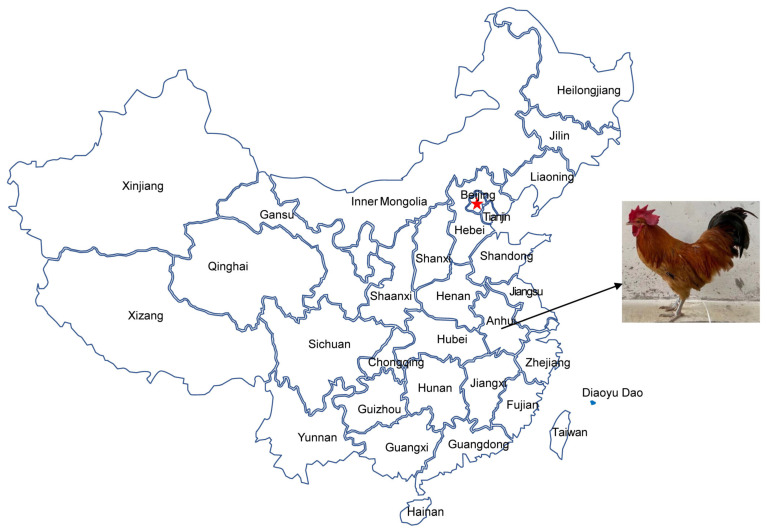
The geographical location of the Pingpu Yellow chicken.

**Figure 2 animals-12-03037-f002:**
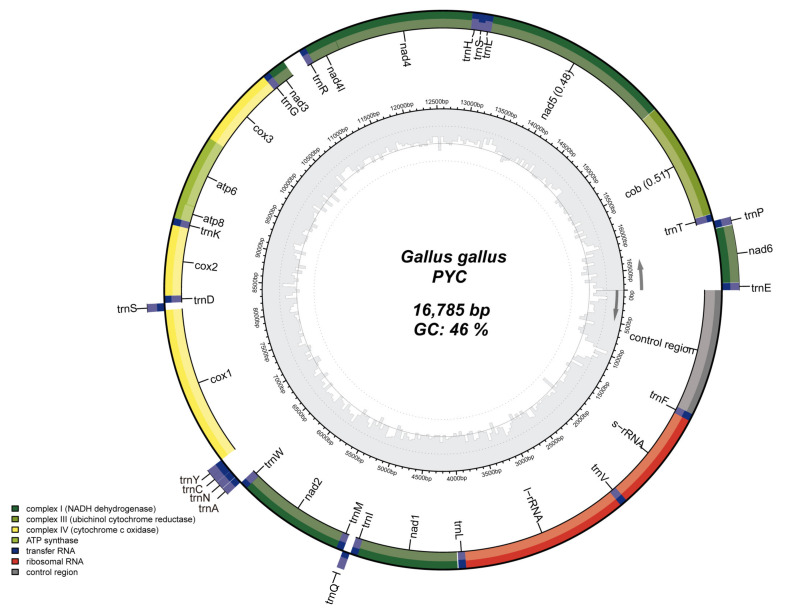
Mitochondrial genome structure of the Pingpu Yellow chicken. Genes in the inner circle plot are located on the N-strand, and the rest of the genes are located on the J-strand. Dark green, light green, and yellow represent PCGs coding complex I, III, and IV, respectively. ATP synthase, rRNA, tRNA genes, and the control region are colored with olive green, red, dark blue, and gray, respectively.

**Figure 3 animals-12-03037-f003:**
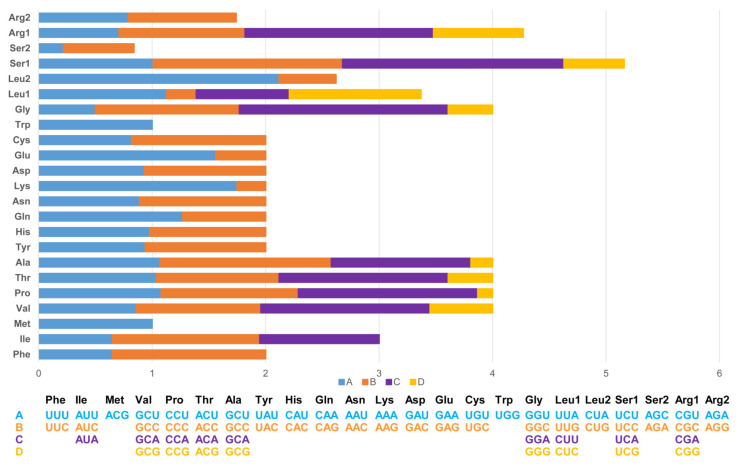
Relative synonymous codon usage (RSCU) in the Pingpu Yellow chicken mitochondrial genome.

**Figure 4 animals-12-03037-f004:**
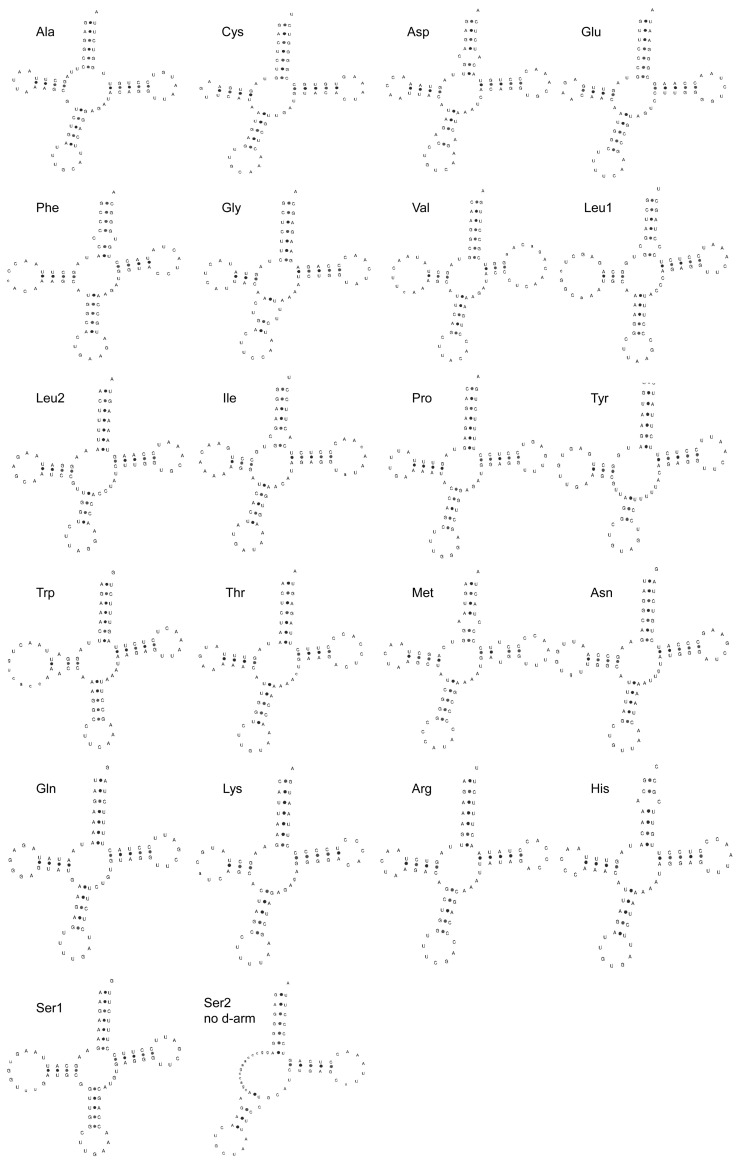
The inferred secondary structure of 22 tRNA genes from the Pingpu Yellow chicken.

**Figure 5 animals-12-03037-f005:**
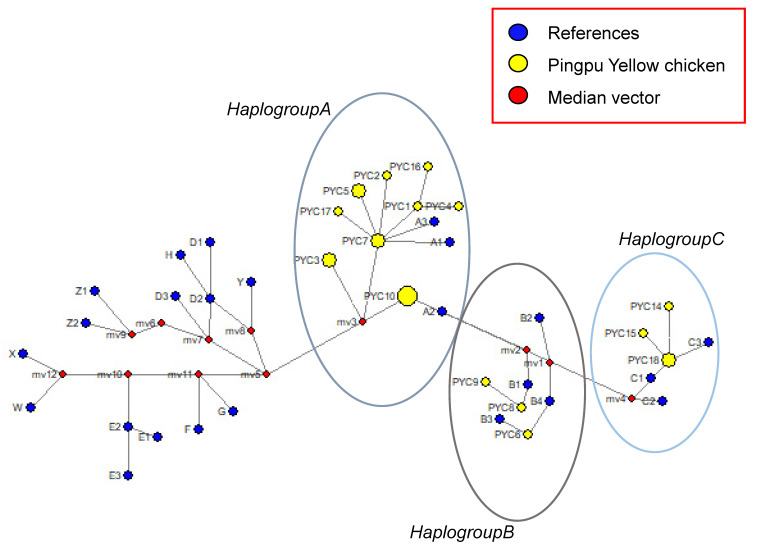
Median-joining network of chicken mtDNA D-loop haplotypes.

**Figure 6 animals-12-03037-f006:**
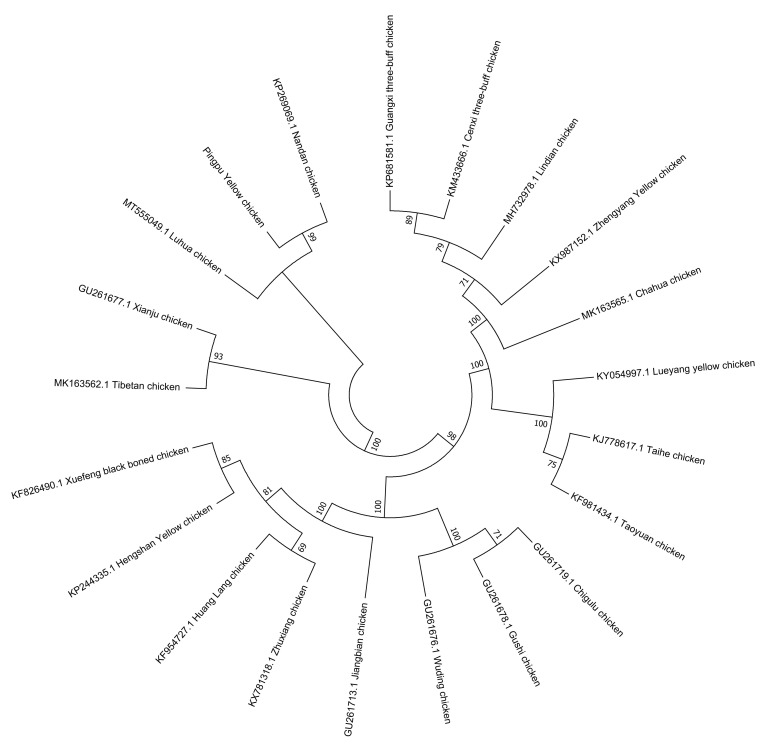
Evolutionary relationships of the Pingpu Yellow chicken and other chicken breeds.

**Table 1 animals-12-03037-t001:** Organization of the Pingpu Yellow chicken mitogenome.

Gene Name	Start and Stop	Length/bp	Strand	Anticodon	Start/Stop Codon
D-loop	1~1231	1231	+	-	-
tRNA-Phe	1232~1301	70	+	GAA	-
s-rRNA	1301~2277	977	+	-	-
tRNA-Val	2277~2349	73	+	TAC	-
l-rRNA	2350~3972	1623	+	-	-
tRNA-Leu	3973~4046	74	+	TAA	-
ND1	4056~5030	975	+	-	ATG/TAA
tRNA-Ile	5031~5102	72	+	GAT	-
tRNA-Gln	5108~5178	71	-	TTG	-
tRNA-Met	5178~5246	69	+	CAT	-
ND2	5247~6287	1041	+	-	ATG/TAG
tRNA-Trp	6286~6361	76	+	TCA	-
tRNA-Ala	6368~6436	69	-	TGC	-
tRNA-Asn	6440~6512	73	-	GTT	-
tRNA-Cys	6514~6579	66	-	GCA	-
tRNA-Tyr	6579~6649	71	-	GTA	-
COX1	6651~8201	1551	+	-	GTG/AGG
tRNA-Ser	8193~8267	75	-	TGA	-
tRNA-Asp	8270~8338	69	+	GTC	-
COX2	8340~9023	684	+	-	ATG/TAA
tRNA-Lys	9025~9092	68	+	TTT	-
ATP8	9094~9258	165	+	-	ATG/TAA
ATP6	9249~9932	684	+	-	ATG/TAA
COX3	9932~10715	784	+	-	ATG/AGA
tRNA-Gly	10716~10784	69	+	TCC	-
ND3	10785~11136	351	+	-	ATG/TAA
tRNA-Arg	11138~11205	68	+	TCG	-
ND4L	11206~11502	297	+	-	ATG/TAA
ND4	11496~12873	1378	+	-	ATG/AGG
tRNA-His	12874~12942	69	+	GTG	-
tRNA-Ser	12943~13009	67	+	GCT	-
tRNA-Leu	13010~13080	71	+	TAG	-
ND5	13081~14898	1818	+	-	ATG/TAA
CYTB	14903~16045	1143	+	-	ATG/TAA
tRNA-Thr	16049~16117	69	+	TGT	-
tRNA-Pro	16118~16187	70	-	TGG	-
ND6	16194~16715	522	-	-	ATG/TAA
tRNA-Glu	16718~16785	1216	-	TTC	-

**Table 2 animals-12-03037-t002:** Nucleotide composition in different regions of the mitochondrial genome of the Pingpu Yellow chicken.

Item	Length(bp)	A%	T%	G%	C%	AT%	GC%	AT-Skew	GC-Skew
Complete mitogenome	16,785	30.3	23.7	13.5	32.5	54.0	46.0	0.1222	−0.4130
PCGs	11,394	29.7	23.3	12.0	35.0	53.0	47.0	0.1208	−0.4894
tRNAs	1547	32.8	24.6	17.1	25.5	57.4	42.6	0.1429	−0.1972
rRNAs	2600	33.0	20.4	18.2	28.4	53.4	46.6	0.2360	−0.2189
Control region	1231	26.6	33.5	13.4	26.6	60.1	40.0	−0.1148	−0.3300

**Table 3 animals-12-03037-t003:** Haplotypes and polymorphic sites of the mtDNA D-loop sequence of the Pingpu Yellow chicken.

Haplotypes	Haplogroup	Localization of Polymorphic Sites																								
		167	212	219	223	225	242	243	246	256	281	296	298	310	315	342	361	363	367	391	396	686	711	792	863	1251
GU261704.1		T	A	C	C	C	G	T	T	T	A	T	C	C	T	A	A	C	T	C	T	G	G	G	A	A
PYC-1	B	*	*	*	T	*	*	*	*	*	*	C	*	*	*	*	*	*	*	*	*	*	*	*	*	*
PYC-2	A	C	G	*	T	T	*	*	C	*	*	C	*	*	C	*	*	*	C	*	*	*	*	*	*	G
PYC-3	C	C	G	*	T	T	*	*	C	C	*	C	*	*	C	*	*	*	*	A	*	*	*	*	*	G
PYC-4	B	*	*	*	T	*	*	*	*	*	*	C	*	*	*	*	*	*	*	*	*	*	*	*	T	*
PYC-5	A	C	G	*	T	T	*	*	C	*	*	C	*	*	C	*	*	*	*	*	C	*	*	*	*	G
PYC-6	B	*	*	*	T	*	*	*	*	*	*	C	*	*	*	*	*	*	*	*	*	*	*	A	*	*
PYC-7	A	C	G	*	T	T	*	*	C	*	*	C	*	*	C	*	*	*	*	*	*	*	*	*	*	G
PYC-8	C	*	*	*	T	*	A	C	C	C	G	C	*	T	C	*	*	T	C	*	*	*	*	*	*	*
PYC-9	C	*	*	*	*	*	A	C	C	C	G	C	*	T	C	*	*	T	C	*	*	*	*	*	*	G
PYC-10	C	*	G	*	T	*	A	C	C	C	G	C	*	T	C	G	*	T	C	*	*	A	*	*	*	G
PYC-11	C	*	G	*	T	*	A	C	C	C	G	C	*	T	C	G	*	T	C	*	*	*	A	*	*	G
PYC-12	B	*	*	*	T	*	*	*	*	*	*	C	T	*	*	*	*	*	*	*	*	*	*	*	*	*
PYC-13	A	C	G	*	T	T	*	*	C	*	*	C	*	*	C	*	*	*	*	*	*	A	*	*	*	G
PYC-14	D	*	G	*	T	*	A	C	C	C	G	C	*	T	C	G	*	T	C	*	*	*	*	*	*	G
PYC-15	A	C	G	T	T	T	*	*	C	*	*	C	*	*	C	*	*	*	*	*	*	*	*	*	*	G
PYC-16	C	*	G	*	T	*	A	C	C	C	G	C	*	T	C	G	*	T	C	T	*	*	*	*	*	*
PYC-17	A	C	G	*	T	T	*	*	C	*	*	C	*	*	C	*	T	*	*	*	*	*	*	*	*	G
PYC-18	C	*	*	*	T	*	A	C	C	C	G	C	*	T	C	G	*	T	C	*	*	*	*	*	*	*

* represents identical nucleotides to the reference sequence (GU261704.1).

## Data Availability

The genome sequence data reported in the article have been deposited in NCBI database (https://www.ncbi.nlm.nih.gov/, accessed on 27 October 2021), which were openly available under the accession number MZ911748.

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
