# Peer review of "Complete Mitochondrial Genome, Genetic Diversity and Phylogenetic Analysis of Pingpu Yellow Chicken (Gallus gallus)"

_animals, 2022, doi:10.3390/ani12213037_

Round 1

Reviewer 1 Report

In this manuscript, Jin and colleagues present a great resource to study mtDNA of Pingpu Yellow chicken bread and investigated their genetic background. The data and the analysis are thorough and high quality. The conclusions are well-supported by the data. I have only a few minor comments:

- Authors may continue their effort to improve language and readability.

- The resolution of the figure must be improved

- Detailed description of the figure legend should be presented, e.g., what’s the light blue represents in the inner circos plot in figure 2? Please check every figure.

- The sequencing data is not available, please make sure to deposit data in public databases

- The title of each session should be a conclusion or description of the finding, e.g., 3.3. Transfer and ribosomal RNA genes, what’s the conclusion of “we found that the nucleotide diversity (Pi) was 0.00600…..” 

- How many female chickens were sequenced?  I guess from the table at the end of MS that is 18, I think it should be mentioned at the beginning  

Author Response

Dear Editor,

The authors thank you and the reviewers for your helpful comments and suggestions on our manuscript, animals-1981967 "Phylogenetic analysis of the complete mitochondrial genome of the Pingpu Yellow chicken". These comments and suggestions have helped us to improve the manuscript. We have made all necessary revisions. The content has been modified to fit the latest Animals format. The addition and changes have been marked in red in the text. There are three major changes in this revision. First, we have provided high resolution figures in the revised manuscript. Second, we have added detailed description of next-generation sequencing of the mitochondrial genome and reduced the repetition rate of revised manuscript. Third, the new references have been cited in the text and the language of re-submitted manuscript was double-checked by a language-editing service.

The responses to reviewers’ comments point-by-point are listed at the end of this letter. We believe that we have carefully addressed all comments and would like to submit the revised manuscript.

Thank you again and look forward to hearing from you soon.

Yours sincerely,  

Zhaoyu Geng, PhD and professor

Department of Animal Genetics and Breeding, College of Animal Science and Technology, Anhui Agricultural University, Hefei 230036, China.

Tel: +86-551-6578 5519

Fax: +86-551-6578 5519

Response to Reviewer 1:

Q1: In this manuscript, Jin and colleagues present a great resource to study mtDNA of Pingpu Yellow chicken bread and investigated their genetic background. The data and the analysis are thorough and high quality. The conclusions are well-supported by the data. I have only a few minor comments: Authors may continue their effort to improve language and readability.

A1: Thanks for your positive comments and constructive suggestions. In our revised manuscript, we have carefully addressed all comments and concerns by Reviewer 1 and made necessary revisions. In addition, we have double-checked our revised manuscript by a language-edited service. Finally, the revised version has been modified to fit the latest Animals format. Please see the revised manuscript highlighted in red.

Q2: The resolution of the figure must be improved.

A2: Done. In the revised manuscript, we have made necessary revisions to improve the resolution of the all figures. Please see the revised manuscript and figures.

Q3: Detailed description of the figure legend should be presented, e.g., what’s the light blue represents in the inner circos plot in figure 2? Please check every figure.

A3: Done. In the figure 2, genes in the inner circle plot are located on the N-strand, and the rest genes on the J-strand. Dark green, light green and yellow represent PCGs coding complex I, III and IV, respectively. ATP syn-thase, rRNA, tRNA genes and the control region are colored with olive green, red, dark blue and gray, respectively.

Q4: The sequencing data is not available, please make sure to deposit data in public databases.

A4: Done. In the present study, the sequencing data were deposited in NCBI database (https://www.ncbi.nlm.nih.gov/), which were openly available under the accession number MZ911748. Please see line 130.

Q5: The title of each session should be a conclusion or description of the finding, e.g., 3.3. Transfer and ribosomal RNA genes, what’s the conclusion of “we found that the nucleotide diversity (Pi) was 0.00600…..”

A5: Done. We have changed the title into “3.3. Transfer and ribosomal RNA genes of PYC”. Please see line 167.

Q6: How many female chickens were sequenced? I guess from the table at the end of MS that is 18, I think it should be mentioned at the beginning.

A6: Done. We have added the detailed information of chicken samples in the “2.2. Specimen collection and DNA extraction”, please see lines 69-73.

Response to Reviewer 2:

Q7: For instance, the title only focuses on the complete mitogenome, and does not mention the second data set/research question.

A7: It is an important advice for improving our revised manuscript. In this study, the complete mitogenome of Pingpu yellow chicken was sequenced using next-generation sequencing. The objectives of the present study were to investigate the complete mitochondrial genome, genetic diversity, and phylogenetic analysis of Pingpu Yellow chicken (PYC). We have changed the title into “Complete mitochondrial genome, genetic diversity and phylogenetic analysis of Pingpu Yellow chicken (Gallus gallus)”. Please see lines 2-3.

Q8: The Simple Summary does not introduce the two research questions at the start but begins with discussing the D-loop. Only a single sentence is devoted to the complete mitochondrial sequence. I recommend that the authors (i) introduce the two research questions at the start, (ii) discuss the mitogenome, and (iii) then discuss the D-Loop findings.

A8: Done. We have added more detailed information according to the useful suggestion. Please see lines 13-15.

Q9: The Abstract does not introduce the two research questions at the start but begins with the mitogenome findings. Halfway the topic shifts to the D-Loop but this is very implicit (“In total, 25 polymorphic sites determined the 18 haplotypes and all major haplogroups (A-C).”). Again I recommend that the authors (i) introduce the two research questions at the start, (ii) discuss the mitogenome, and (iii) then discuss the D-Loop findings, but make it very clear that this is a different data set. The D-Loop findings should be clearly announced, so e.g. “In the D-Loop data set, we found 25 polymorphic sites which determined 18 haplotypes and three major haplogroups”.

A9: Done. We have made revisions. Please see lines 31-32.

Q10: Nowhere in the abstract is it made clear how many chickens were sequenced. This should be mentioned.

A10: Done. We have added the detailed number of samples in the revised abstract, please see lines 23-25.

Q11: Line 23: change “size” to “total length”.

A11: Done . Please see line 25.

Q12: Line 31: Change “indicating” to “indicates”.

A12: Done. Please see line 35.

Q13: Line 55: Please indicate the sample size.

A13: Done. We have added the number of samples in the section “2.2. Specimen collection and DNA extraction”, please see lines 69-73.

Q14: Lines 64-66: this does not belong under the heading “Specimen collection and DNA extraction” but should be a separate heading: 2.3 Next-generation sequencing of the mitogenome. This is too brief. Please indicate how the mitogenome was assembled (de novo or using a reference mitogenome, and if so, which one), which program was used to assemble the mitogenome, what was the coverage of the mitogenome, etc.

A14: Done. We have added a separate heading about next-generation sequencing of the mitogenome. Please see lines 87-92.

Q15: Line 67: indicate in the heading that this is about the D-Loop

A15: Done. We have changed the heading into “2.3. PCR amplification and DNA sequencing of D-loop”. Please see line 75.

Q16: Line 78: this section should start with the mitogenome data, followed by the D-Loop data. These should be separate paragraphs under 2.4.

A16: Done. We have made revisions and divided different paragraphs. Please see lines 93-125.

Q17: Line 100 and further: please indicate the outgroup. Which outgroup sequence was selected, and why this one. Another species of Gallus would be best (unequivocal). Did you used the entire mitogenomes for phylogenetic analysis of just the PCGs?

A17: A good question. We downloaded 20 chicken breed reference sequences from NCBI database to construct neighbor-joining (NJ) tree. Additionally, the whole mitochondrial genome sequences of local chicken breeds were selected according to different breed features, distribution, and regional characteristics. Please see line 103.

Q18: Lines 114-116: Here it is stated that four protein-coding genes were located on the negative strand and that nine protein-coding genes were located on the positive strand. This is incorrect. See Table 1, where it is stated that all but one (ND6) of the protein-coding genes were on the positive strand. Only ND6 is on the negative strand.

A18: Done. Please see Table 1.

Q19: Line 124: Change “Condon” to “Codon”

A19: Done. Please see line 148.

Q20: Line 144: “The secondary structure of tRNAs is inferred by the tRNAscan-SE v2.0 [26].” This is a statement about the methods used and thus belongs in the methods section.

A20: Done. We have moved the sentence into the section of Materials and methods (2.5. Data analysis). Please see lines 102-103.

Q21: Line 178: Change “Bule” to “Blue”

A21: Done. We have changed Bule into Blue. Please see line 200.

Q22: Line 180: please indicate in the header that the analysis deals with mitogenomes

A22: Done. We have changed the header into “3.5 Phylogenetic reconstructions based on complete mtDNA”. Please see line 203.

Q23: Line 202: Gu and Li [23] and Miao et al. [22]

A23: Done. We have modified the format of references. Please see line 225.

Q24: Line 206: delete “the PYC was clustered with”

A24: Done. Please see line 228.

Reviewer 2 Report

This is a fine descriptive study of the mitogenome of the Pingpu Yellow Chicken, and the intra-breed genetic diversity of this breed.

My comments are relatively minor and are easy to fix.

The study uses two data sets, which were obtained with different sequencing methods, and which address two different research questions: (i) the complete mitogenome of a single bird was sequenced using next-generation sequencing methodology and was used to assess the phylogenetic position of the Pingpu Yellow Chicken; (ii) the D-Loop of multiple birds was sequenced using traditional primer-based Sanger sequecing and was used to address intra-breed genetic diversity/differentiation.

This is not always clear in the manuscript.

For instance, the title only focuses on the complete mitogenome, and does not mention the second data set/research question.

The Simple Summary does not introduce the two research questions at the start but begins with discussing the D-loop. Only a single sentence is devoted to the complete mitochondrial sequence. I recommend that the authors (i) introduce the two research questions at the start, (ii) discuss the mitogenome, and (iii) then discuss the D-Loop findings.

The Abstract does not introduce the two research questions at the start but begins with the mitogenome findings. Halfway the topic shifts to the D-Loop but this is very implicit (“In total, 25 polymorphic sites determined the 18 haplotypes and all major haplogroups (A-C).”). Again I recommend that the authors (i) introduce the two research questions at the start, (ii) discuss the mitogenome, and (iii) then discuss the D-Loop findings, but make it very clear that this is a different data set. The D-Loop findings should be clearly announced, so e.g. “In the D-Loop data set, we found 25 polymorphic sites which determined 18 haplotypes and three major haplogroups”.

Nowhere in the abstract is it made clear how many chickens were sequenced. This should be mentioned.

Line 23: change “size” to “total length”

Line 31: Change “indicating” to “indicates”

Line 55: Please indicate the sample size.

Lines 64-66: this does not belong under the heading “Specimen collection and DNA extraction” but should be a separate heading: 2.3 Next-generation sequencing of the mitogenome

Lines 64-66: This is too brief. Please indicate how the mitogenome was assembled (de novo or using a reference mitogenome, and if so, which one), which program was used to assemble the mitogenome, what was the coverage of the mitogenome, etc.

Line 67: indicate in the heading that this is about the D-Loop

Line 78: this section should start with the mitogenome data, followed by the D-Loop data. These should be separate paragraphs under 2.4.

Line 100 and further: please indicate the outgroup. Which outgroup sequence was selected, and why this one. Another species of Gallus would be best (unequivocal).

Line 100 and further: Did you used the entire mitogenomes for phylogenetic analysis of just the PCGs?

Lines 114-116: Here it is stated that four protein-coding genes were located on the negative strand and that nine protein-coding genes were located on the positive strand. This is incorrect. See Table 1, where it is stated that all but one (ND6) of the protein-coding genes were on the positive strand. Only ND6 is on the negative strand.

Line 124: Change “Condon” to “Codon”

Line 144: “The secondary structure of tRNAs is inferred by the tRNAscan-SE v2.0[26].” This is a statement about the methods used and thus belongs in the methods section.

Line 178: Change “Bule” to “Blue”

Line 180: please indicate in the header that the analysis deals with mitogenomes

Line 202: Gu and Li [23] and Miao et al. [22]

Line 206: delete “the PYC was clustered with”

Author Response

(The authors gave the same response as above.)
